# In Vitro and In Silico Studies on Cytotoxic Properties of Oxythiamine and 2′-Methylthiamine

**DOI:** 10.3390/ijms25084359

**Published:** 2024-04-15

**Authors:** Marta Malinowska, Magdalena Czerniecka, Izabella Jastrzebska, Artur Ratkiewicz, Adam Tylicki, Natalia Wawrusiewicz-Kurylonek

**Affiliations:** 1Faculty of Chemistry, University of Bialystok, Ciolkowskiego 1K, 15-245 Bialystok, Poland; m.malinowska@uwb.edu.pl (M.M.); gierdas@uwb.edu.pl (I.J.); 2Faculty of Biology, University of Bialystok, Ciolkowskiego 1J, 15-245 Bialystok, Poland; atyl@uwb.edu.pl; 3Laboratory of Tissue Culture, Department of Biology, University of Bialystok, Ciolkowskiego 1J, 15-245 Bialystok, Poland; 4Department of Clinical Genetics, Medical University of Białystok, Waszyngtona 13, 15-089 Bialystok, Poland; natalia.wawrusiewicz-kurylonek@umb.edu.pl

**Keywords:** thiamine antimetabolites, cell growth inhibition, molecular docking, molecular dynamics, OCT1

## Abstract

It is important to search for cytostatic compounds in order to fight cancer. One of them could be 2′-methylthiamine, which is a thiamine antimetabolite with an additional methyl group at the C-2 carbon of thiazole. So far, the cytostatic potential of 2′-methylthiamine has not been studied. We have come forward with a simplified method of synthesis using commercially available substrates and presented a comparison of its effects, as boosted by oxythiamine, on normal skin fibroblasts and HeLa cancer cells, having adopted in vitro culture techniques. Oxythiamine has been found to inhibit the growth and metabolism of cancer cells significantly better than 2′-methylthiamine (GI_50_ 36 and 107 µM, respectively), while 2′-methylthiamine is more selective for cancer cells than oxythiamine (SI = 180 and 153, respectively). Docking analyses have revealed that 2′-methylthiamine (Δ*G* −8.2 kcal/mol) demonstrates a better affinity with thiamine pyrophosphokinase than thiamine (Δ*G* −7.5 kcal/mol ) and oxythiamine (Δ*G* −7.0 kcal/mol), which includes 2′-methylthiamine as a potential cytostatic. Our results suggest that the limited effect of 2′-methylthiamine on HeLa arises from the related arduous transport as compared to oxythiamine. Given that 2′-methylthiamine may possibly inhibit thiamine pyrophosphokinase, it could once again be considered a potential cytostatic. Thus, research should be carried out in order to find the best way to improve the transport of 2′-methylthiamine into cells, which may trigger its cytostatic properties.

## 1. Introduction

Thiamine (vitamin B1, Figure 1) is one of the key vitamins necessary for the proper functioning of all cells and organisms. Its involvement in metabolism mainly impacts basic bioenergetic pathways, where, after phosphorylation into thiamine pyrophosphate, it is the coenzyme of the pyruvate dehydrogenase complex, catalyzing the oxidative decarboxylation of pyruvate, and thus contributing to the formation of the acetyl-Co-A and 2-oxoglutarate dehydrogenase complex, being one of the main regulatory enzymes of the Krebs cycle. In addition, thiamine pyrophosphate is the coenzyme of transketolase, that is an important enzyme of the pentose–phosphate pathway which generates the NADPH required for the synthesis of fatty acids, steroids, nucleotides, and many other organic compounds [1]. The essence of the thiamine pyrophosphate coenzymatic action derives from the deprotonation of the C-2 carbon of the thiazole ring with ylide formation [2,3], which facilitates the decarboxylation process of especially small molecules like 2-oxoacids (pyruvate- and 2-oxoglutarate dehydrogenases) [4,5,6] or the transfer of two-carbon fragments from the donor to the acceptor, especially in the metabolism of sugars (transketolase) [7,8].

Due to its involvement in basic metabolic processes and potential practical use in research and clinical practice, the synthesis of thiamine derivatives is of constant interest in biology and medicine. Some of them have found applications in modeling thiamine-deficient states and elucidating the mechanism of thiamine coenzymatic action. Others are used in human and veterinary medicine as effective drugs in infections with various microorganisms, or help to improve thiamine deficiency states [9,10,11]. Of particular interest to researchers are thiamine antimetabolites that show cytostatic and antimicrobial properties. Due to the high tolerance of some thiamine antimetabolites (oxythiamine, benfooxythiamine) by human healthy cells (skin fibroblasts) [12], their use in the treatment of lung cancer [13,14], pancreatic cancer [15,16], Ehrlich’s tumor [17], fungal infections [18], and even viral infections [19] is postulated. Commercially available oxythiamine (Figure 1), that is characterized by modification in the pyrimidine ring (it is an amine group instead of a hydroxyl group) is one of the most potent and well-studied inhibitors of thiamine pyrophosphate-dependent enzymes [10]. Research is currently underway to synthesize thiamine analogues within the thiazole ring structure to enhance cytostatic effects [20].

One thiamine analogue with an altered thiazole ring structure is 2′-methylthiamine (Figure 1). In this derivative, the C-2 carbon of the thiazole ring is substituted with a methyl group, which prevents its deprotonation, so the coenzymatic activity of this derivative should be completely canceled. Given the chemical structure of 2′-methylthiamine and the mechanism of thiamine diphosphate-dependent reactions, this analogue meets all the requirements of a potent thiamine antimetabolite. Although the route of 2′-methylthiamine synthesis has been known since the second half of the last century [21], its impact upon cells is practically unknown. The literature only mentions the lack of coenzymatic activity of 2′-methylthiamine in the in vitro studies of thiamine-dependent pyruvate decarboxylase isolated from yeast [21]. This fact does not conclusively determine the cytostatic properties of the aforementioned thiamine derivative against human cells but is a premise for the occurrence of those features. The lack of data on the effects of 2′-methylthiamine on human cells may be due to the complicated route of synthesis and, consequently, the availability restrictions of the compound for biological studies.

Knowing the structure of 2′-methylthiamine, and the role of deprotonation of the C-2 carbon of the thiazole ring of thiamine in the catalytic processes, we hypothesize that 2′-methylthiamine, due to its similar structure to thiamine and inability to form ylide, will be a potent inhibitor of thiamine-dependent processes in cells. Thus, 2′-methylthiamine should show equally strong, if not stronger, cytostatic properties against cancer cells as compared to oxythiamine. Our research aims to synthesize 2′-methylthiamine from commercially available substrates by means of a new, simplified method, and verify the above hypothesis by comparing the cytostatic effects of oxythiamine and 2′-methylthiamine in in vitro cultures of HeLa cells and normal skin fibroblasts. To propose a mechanism of 2′-methylthiamine action and compare it with oxythiamine action, we have used the currently available tools of computational chemistry, such as molecular docking and molecular dynamics. In order to predict possible interactions between the tested ligands and the thiamine diphosphate-dependent enzymes as well as thiamine transporter, models of selected proteins that are available in the Protein Data Bank have been used. The use of theoretical chemistry tools together with the results of in vitro experiments has allowed us to explain the observed effects and propose solutions that could support future research on 2’-methylthiamine as a potential cytostatic. Moreover, by shedding a new light on the mechanism of action of thiamine derivatives and the role of their transport into cells, we show the research objectives that will indicate new possibilities for the use of already known antimetabolites of thiamine.

## 2. Results and Discussion

### 2.1. Synthesis of 2′-Methylthiamine

We have developed an improved method for 2′-methylthiamine preparation (Figure 1) based on the modified synthesis previously published by Bag et al. [22]. First, the starting material, 4-amino-2-methylpyrimidine-5-carbonitrile (**1**) was synthesized by reacting acetamidine and ethoxymethylenemalononitrile [23] to obtain 68% yield. That process was followed by the reduction to 4-amino-2-methylpyrimidine-5-carbaldehyde (**2**) in the presence of Raney Nickel in 80% formic acid with the subsequent reaction to (4-amino-2-methylpyrimidin-5-yl)methanol (**3**) by treatment with NaBH_4_ (68% yield after two steps). Next, treating alcohol **3** by HBr in glacial acetic acid [24] resulted in bromide hydrobromide salt, which was followed by the next reaction without product characterization. In parallel, 2,4-dimethyl-5-(2-hydroxyethyl)thiazole (**4**) was obtained according to the literature-based procedure [25,26,27]. Taking bromide and **4** in hand, the final step was a condensation reaction in excess of thiazole to give the desired 2′-methylthiamine in 20% yield.

### 2.2. The Effects of Oxythiamine and 2′-Methylthiamine on Cells In Vitro

Microscopic observations of in vitro cultures indicate that both oxythiamine and 2′-methylthiamine inhibit the growth of HeLa tumor cells, with oxythiamine having a stronger effect. We did not find a similar effect in the case of the fibroblasts and thiamine cultures (Figure 2). In order to better explore the observed effect, the curves of cell growth as a function of antivitamin concentration were performed (Figure 3A,B,D,E,G,H). Data were collected when confluence was reached in the control cultures.

We did not observe any differences in the growth of cells cultured in the thiamine-supplemented medium (Figure 3A,B), although data from the literature suggest that thiamine may reduce tumor cell proliferation [28]. However, these used concentrations beyond the range tested in our study. Oxythiamine at the concentration of 47 µM caused a higher than five-fold decrease in the HeLa cell number as compared to the control (Figure 3G,H), while 2′-methylthiamine at the same concentration caused a decrease in the cell number by only about a half (Figure 3D,E). At the maximum concentration of the tested antivitamins (1500 µM), the number of 2′-methylthiamine-treated HeLa cells approximately equaled 25% of the control, while that of the oxythiamine-treated cells dropped to just 2% of the control. In the case of fibroblasts, we did not find any significant differences in the cell numbers in comparison to the controls in the case of both 2′-methylthiamine and oxythiamine-treated cultures (Figure 3D,E,G,H). These results indicate a limited sensitivity of healthy cells to thiamine antimetabolites in contrast to cancer cells. The results from the previous studies [12] where human skin fibroblasts were treated with oxythiamine have indicated that, in addition to not affecting cell survival in the concentration range from 30 µM to 1000 µM, oxythiamine stimulates collagen synthesis by activating prolidase and the factors regulating the expression of genes associated with collagen synthesis. Similar results were obtained in the studies on the effects of benfooxythiamine (a novel prodrug releasing oxythiamine) and oxythiamine on primary human gingival fibroblasts, where it was shown that both compounds caused only a slight reduction in cell viability (about 10%) at the concentration of 100 µM [19]. On the contrary, in the above-mentioned experiments, oxythiamine reduced the proliferation of cancer cells [12,19] and tumor growth in models of various cancers [15,17]. Oxythiamine is one of the more extensively studied thiamine antimetabolites [10]. Its effects on metabolism have been studied at different levels as follows: in vivo animal studies, and in vitro on various cell lines and isolated thiamine diphosphate-dependent enzymes. It has been shown that thiamine antivitamins may act as substrates for thiamine pyrophosphokinase to form their pyrophosphate esters [29,30]. That process may impact the thiamine pyrophosphate synthesis and provide for an anticoenzyme that has a high affinity with thiamine pyrophosphate-dependent enzymes, including transketolase, the activity of which is very important for cancer cells [31,32,33]. The IC_50_ of oxythiamine for rat liver transketolase was 0.2 μM [34] and for yeast, transketolase was approximately 0.03 μM, and even the addition of 0.5 μM of thiamine diphosphate did not restore enzyme activity [35]. The cited data indicate that oxythiamine may be a very good inhibitor of transketolase. Unfortunately, in the related literature, there is no data on the interaction of 2′-methylthiamine with this enzyme. The only information about the interaction of 2′-methylthiamine with thiamine-dependent enzymes concerns pyruvate decarboxylase derived from yeast [21] since that derivative has not been proven to have catalytic properties comparable to that of thiamine.

Measurement of the changes in the metabolic activity of the HeLa cells and fibroblasts (MTT assay) have shown the metabolic activity of HeLa cells to have decreased significantly under the influence of oxythiamine at the concentration of 47 µM (50% of control, Figure 3I), while for HeLa cells treated with 2′-methylthiamine, a statistically significant decrease of 30% concerning the control was observed at the concentration of 750 µM (Figure 3F). In contrast to the HeLa cells, we did not find any significant differences in the metabolic rate of fibroblasts as compared to the control for both thiamine antivitamins tested (Figure 3F,I). In addition, we observed that the metabolic activity of fibroblasts slightly increased under the influence of thiamine at concentrations above 47 µM (Figure 3C). In order to make a direct comparison of the effects of thiamine and the tested thiamine antivitamins on HeLa cells and fibroblasts, we determined the values of the GI_50_, IC_50_, and SI coefficients (Table 1). The obtained GI_50_ and IC_50_ values for the tested thiamine derivatives confirmed previous observations. The average GI_50_ (39 µM) and IC_50_ (51 µM) values for oxythiamine are approximately two times lower than those reported for 2′-methylthiamine (83 µM and 112 µM, respectively) in the case of HeLa cells, while the above-mentioned coefficients for thiamine and both tested thiamine derivatives for fibroblasts go substantially beyond the range of the tested concentrations.

Summarizing the experimental data obtained, it must be concluded that contrary to the initial assumption, oxythiamine shows a much stronger cytostatic effect on tumor cells as compared to 2′-methylthiamine, and that normal human fibroblasts are resistant to the negative effects of both thiamine derivatives tested. However, attention should be drawn to the SI (Table 1) that is significantly higher (by about 20%) for 2′-methylthiamine. It demonstrates the better selectivity of its action against tumor cells as compared to oxythiamine. The data also further indicate the potential of 2’-methylthiamine as a cytostatic.

Analyzing the obtained results in relation to the chemical structure of the antivitamins studied, we have hypothesized that the differences in their effect on cancer cells might be due to the differences in their affinity with enzymes or proteins involved in the thiamine transport. In order to verify this hypothesis in further studies, we have used the computational chemistry tools to compare the interaction of the derivatives tested with the thiamine diphosphate-dependent enzymes and proteins responsible for the thiamine transport.

### 2.3. Molecular Docking

For a comprehensive analysis, three enzymes (transketolase (EC 2.2.1.1), pyruvate dehydrogenase (EC 1.2.4.1), and thiamine pyrophosphokinase EC 2.7.6.2)) have been taken into account, employing computational methods to shed light on the observed disparities in the experimental data.

#### 2.3.1. Docking to Enzymes—Affinities

The preliminary docking comparisons of both thiamine derivatives, as pyrophosphate forms, and the natural coenzyme to three tested enzymes revealed a huge variation of matches. In order to facilitate the data analysis, we have grouped those matches into similar positions (Table 2). In the case of transketolase and pyruvate dehydrogenase, we have identified two positions for each of the three ligands. In the case of thiamine pyrophosphokinase, we found three positions for each ligand. The comparison of the recorded ligand positions in the active centers of the enzymes is shown in Figure 4. For all the analyzed enzymes, the first of the identified positions has the highest number of single matches (Table 2), and that position also contains the match with the lowest docking affinity (Table 3). When comparing the mean docking affinity determined for position 1 for transketolase, we found no difference between thiamine pyrophosphate and oxythiamine pyrophosphate, at −5.5 kcal/mol (71.9% matches) and −5.6 kcal/mol (65.7% of all the matches), respectively. However, 2′-methylthiamine pyrophosphate had a higher mean docking affinity as compared to thiamine pyrophosphate at −5.3 kcal/mol (71.7% of all the matches). Similar relationships can be seen for the lowest recorded docking affinity for a single match at −6.2 kcal/mol, −6.2 kcal/mol, and −6.1 kcal/mol for thiamine pyrophosphate, oxythiamine pyrophosphate, and 2′-methylthiamine pyrophosphate, respectively (Table 3). In the case of pyruvate dehydrogenase, similar relationships have been observed. Over 99% of the matches found for each compound exhibit the lowest average docking affinity. For thiamine and oxythiamine pyrophosphate, it is −9.1 kcal/mol, while for 2′-methylthiamine pyrophosphate, it is −8.3 kcal/mol. The minimum docking affinity is −9.8 kcal/mol for thiamine pyrophosphate, −9.5 kcal/mol for oxythiamine pyrophosphate, and −9.2 kcal/mol for 2′-methylthiamine pyrophosphate. A different relationship has been observed for thiamine pyrophosphokinase, where the average docking affinity for position 1 is the lowest at −7.5 kcal/mol (as well as the lowest docking affinity noted for a single match at −8.2 kcal/mol) for 2′-methylthiamine than that recorded for thiamine, while the docking affinities for oxythiamine and thiamine were similar (−7.0 kcal/mol and 7.1 kcal/mol, respectively, Table 3).

Taking into account the above data, it is plausible to conclude that the binding of the anticoenzyme (oxythiamine pyrophosphate) in the active sites of transketolase and pyruvate dehydrogenase is competitive and possibly the same as the binding of thiamine pyrophosphate. In addition, the docking analyses of those ligands at the active centers of both enzymes indicate a very similar localization when comparing the most common position with the most convenient average docking affinity (Figure 5). In contrast, the binding of 2′-methylthiamine pyrophosphate in the active sites of those two enzymes may be more difficult than in the case of a native coenzyme because of the higher average docking affinity as compared to thiamine pyrophosphate.

#### 2.3.2. Docking to Enzymes—Ligand Arrangements

In addition, the arrangement of 2′-methylthiamine pyrophosphate in the active center of both enzymes has been found to differ significantly from that of the native coenzyme (Figure 6). This observation would confirm the effects noted as a result of the experiment on HeLa cells, where oxythiamine shows a stronger cytostatic effect as compared to 2′-methylthiamine if it were not for the fact that 2′-methylthiamine shows a lower docking affinity with thiamine pyrophosphokinase. Thiamine pyrophosphokinase is the main enzyme supplying cells with the coenzymatically active form of vitamin B1 and has a very high affinity with it, having great importance for thiamine uptake and distribution in cells, as well as the activity of all thiamine pyrophosphate-dependent enzymes [36,37,38]. Although the oxythiamine and thiamine docking affinities are almost identical (Table 3), the difference in the forming interactions with the active center of the enzyme may be more favorable in the case of thiamine. As shown in Figure 6, oxythiamine is involved in the adverse acceptor–acceptor interactions with ASP 100, which may indicate the presence of repulsive forces between ligand and target. According to our results, the probability of 2′-methylthiamine binding in the active site of thiamine pyrophosphokinase is higher than in the native substrate (Table 3). Moreover, the localization of 2′-methylthiamine in the active center of thiamine pyrophosphokinase is in the opposite conformation as compared to thiamine (Figure 5). The incorrect location of the substrate hydroxyl group in the active center of the enzyme may prevent the synthesis of the thiamine pyrophosphate ester. Our ligand-binding analyses of the 2′-methylthiamine binding with the enzyme’s active center indicate that the amino group of the pyrimidine ring establishes five additional ionic interactions (including the salt bridge) that are absent in the case of oxythiamine and thiamine (except for the ASP 100 residue, Figure 6).

This explains the lower docking affinity with 2′-methylthiamine relative to the other ligands and indicates that 2′-methylthiamine may strongly inhibit the enzyme. Thus, it is plausible to assume that if 2′-methylthiamine enters the cells; it should significantly inhibit the activity of thiamine pyrophosphokinase and thus limit the availability of the active coenzyme for all the enzymes dependent on thiamine pyrophosphate, thereby limiting their activity as well.

#### 2.3.3. Docking to Transporters

Considering the above interpretation of the docking results, the question of why 2′-methylthiamine has a much weaker effect on cancer cells as compared to oxythiamine remains open. Searching for an answer to that question, we decided to investigate the possible effects of the tested thiamine antimetabolites on membrane thiamine transporters. Three proteins in the SLC19A (solute carrier transporters) family, THTR1, THTR2, and RFC1, are involved in the transport of thiamine into mammalian cells. The THTR1 (SLC19A2) and THTR2 (SLC19A3) transporters act as antiporters of thiamine and hydrogen ions. The two transporters differ significantly in their affinity with thiamine. The THTR1 transporter (Km = 2.5 µM) has a significantly lower affinity with thiamine compared to the THTR2 transporter (Km = 2.7 × 10^−4^ µM) [37,39,40]. Another RFC1 protein (SCL19A1), a reduced folate transporter, does not transport thiamine but only the phosphate derivatives of that vitamin. Likely through its involvement in pumping thiamine mono- and diphosphate out of the cell, it contributes to the regulation of intracellular levels of the thiamine phosphate esters (in both cofactor and non-cofactor forms) [39,41]. Although the function of these proteins as well as the sequence of the genes encoding them are well understood, their crystallographic data in terms of the PDB are not available. Therefore, in our study, we have been forced to use another protein involved in transporting thiamine into cells, the crystal structure of which is available in the Protein Data Bank, the human organic cation transporter OCT1 (gene SLC22A1) and ThiT. Human OCT1 is located mainly in hepatocytes, enterocytes, and renal cells, where it mediates the facilitated transport of a variety of organic cations, including thiamine [42]. The importance of the OCT1 transporter for cellular thiamine levels is described in the mice model [43,44]. In humans, the affinity of OCT1 with thiamine is lower than that of THTR1 and THTR2; however, the Vmax of OCT1 is significantly higher in comparison to THTR1 and THTR2 [45]. It may therefore be assumed that OCT1 plays a role in transporting thiamine into human cells. The results of the recent study indicate that they may be related in transport of drugs to cancer cells (Km for doxorubicin 4.6 µM). Moreover, the expression of OCT1 is elevated in tumorous breast tissue as compared to normal breast tissue [46]. Other data indicate that the OCT1 expression in tumor tissue is lower than in surrounding healthy tissue, particularly in the advanced stages of cancer [47,48]. Due to elevated metabolism, the tumor tissues are more sensitive in the case of reduced thiamine availability. That may explain the lower cytotoxic effect of thiamine antimetabolites on tumor cells as compared to fibroblasts. On the other hand, the enhanced expression of OCT1 is correlated with less differentiation toward the kidney tumor cells and is the positive prognostic value in the case of various liver cancers due to involvement of this protein in cytostatics transport [47,48,49]. ThiT is the S-component of the thiamine-specific energy coupling factor ECF derived from *Lactococcus lactis*, which is a known prokaryotic thiamine transporter that has been used in the studies of the transport of various thiamine derivatives into bacterial cells [50,51,52].

#### 2.3.4. Docking Statistics

A general analysis of thiamine, oxythiamine, and 2′-methylthiamine docking in relation to the transport proteins (Table 4) indicates that the 2′-methylthiamine or thiamine matches with OCT1 are grouped into three or four positions, while 2′-methylthiamine is characterized by only one position for the ThiT transporter. Oxythiamine interacted with ThiT in two positions while it interacted with OTC1 in four positions. We have found the greatest variation in the docking capabilities of all ligands for OCT1, where the individual matches have been grouped into four positions in the case of thiamine as well as oxythiamine, and three positions for 2′-methylthiamine (Table 4 and Figure 7). The analysis of the docking affinity of the individual ligands with both thiamine transporters has shown that the lowest docking affinities have been recorded for a single match assigned to positions with the highest number of matches. Those positions have also exhibited the lowest average docking affinity (Table 5). Hence, the most likely docking position for thiamine is position 2 for OCT1 (average Δ*G* −6.5 kcal/mol, 72.3% of all the matches) as well as ThiT (average Δ*G* −7.8 kcal/mol, 77.8% of all the matches), position 2 for ThiT (average Δ*G* −7.9 kcal/mol, 53.3% of all the matches) and position 1 for OCT1 (average Δ*G* −6.5 kcal/mol, 61.8% of all the matches) for oxythiamine, and position 3 for OCT1 (average Δ*G* −7.1kcal/mol, 70.1% of all the matches) and position 1 for ThiT (average Δ*G* −7.0 kcal/mol, 100.0% of all the matches) for 2′-methylthiamine. Given the data on the docking affinity of the individual ligands with thiamine transporters, it should be concluded that oxythiamine can be incorporated into both transporters with the similar probability as thiamine due to the lack of significant differences in the docking affinity for both ligands (Table 5). Taking into consideration the similar localization of thiamine and oxythiamine in the most beneficial position in ThiT as well as OTC1 molecules (Figure 8), it can be assumed that oxythiamine will be transported into human as well as bacterial cells, where it can be phosphorylated and inhibit, in the form of pyrophosphate, the activity of thiamine pyrophosphate-dependent enzymes thus causing a reduction of the cell growth rate.

#### 2.3.5. Docking—Discussion

This section interprets the outcomes of our HeLa cell experiment. The highlighted observation holds particular significance for prokaryotic cells, hinting at the potential utility of oxythiamine as an antibiotic. Yet, our search has yielded limited data on the oxythiamine–bacteria interaction [53]. In the case of 2′-methylthiamine, we show different docking affinity as compared to thiamine (Table 5). In the case of bacterial transporter ThiT, 2′-methylthiamine shows a significantly higher docking affinity as compared to thiamine, which indicates the high specificity of that transporter to thiamine and the low probability of 2′-methylthiamine binding. In addition, the pattern of 2′-methylthiamine docking with ThiT in the most optimal position indicates the opposite orientation of the ligand in the protein molecule as compared to the docking of thiamine (Figure 7), which may adversely impact the transport rate. Thus, it can be assumed that 2′-methylthiamine will not be transported into bacterial cells, so there is no basis for considering it as an antibiotic. On the contrary, in the case of human OCT1, 2′-methylthiamine shows a significantly lower docking affinity as compared to thiamine and oxythiamine, so it can be predicted that it will be more easily bound by this transporter than thiamine. Additionally, the inverted location of the thiazole ring compared to the docking of thiamine and oxythiamine (Figure 8) may hinder, if not prevent, the transport of 2′-methylthiamine into cells. These assumptions are supported by the analysis of intermolecular interactions between thiamine, oxythiamine, and 2′-methylthiamine, and the OCT1 transporter proteins (Figure 9). This binding analysis has demonstrated that thiamine forms a stable conventional hydrogen bond with residues SER 470 and GLN 241. Those interactions are further stabilized by the presence of four alkyl/π-alkyl interactions between the pyrimidine ring, and MET 218, CYS 450, TRP 354, and LYS 214. Thiamine, as a cation, forms a strong ionic bond (charge–charge interactions) with ASP 474, as well as extra π-cation interactions with the residue TYR 36. Moreover, thiamine has also exhibited π–π stacked interactions with TYR 36 at the distance of 5.16 Å, and π-sigma at PHE 244 at 3.64 Å. Oxythiamine binds via identical interactions as thiamine except for the presence of an additional hydrogen bond with LYS 214, and the absence of a strong charge–charge electrostatic attraction with ASP 474. The consequence of introducing an additional methyl group to form 2′-methylthiamine is the rotation around the single bond between the methylene bridge and thiazolium ring, hence it has a different arrangement in the transporter canal. That ligand interacts with ASP 474 by the strong ionic bond, just like in the case of thiamine, but the additional methyl moiety forms π-sigma interactions with TRP 217 as well as PHE 244 (instead of 4′-methyl group) at the distances of 3.96 Å and 3.73 Å, respectively. To sum up, it should be emphasized that 2′-methylthiamine is extra stabilized by interactions with a shorter distance, and therefore it binds stronger than thiamine and oxythiamine. The inability of 2′-methylthiamine to be transported into the cells may probably be the main reason for its weaker effect on cancer cells as compared to that of oxythiamine, which is transported into the cells as evidenced by the results of other research on rats and yeast [54,55] (Figure 2 and Table 2).

#### 2.3.6. Docking—Future Directions

Since the experiment demonstrates the significant anti-tumor activity of 2′-methylthiamine, it is interesting to investigate the derivatives with similar effectiveness but improved penetration through relevant transporters. Specifically, replacing one of 2′-methylthiamines’s methyl groups with another could potentially combine its anticancer properties with enhanced transport capabilities. However, given the numerous potential substituent combinations, it is essential to conduct prior in silico verification of prospective compounds. QSAR-type studies will also be beneficial in preliminarily assessing their suitability as drug candidates.

### 2.4. Molecular Dynamics

Molecular docking simulations suggest a weakening of 2’-methylthiamine due to steric effects caused by an additional methyl group. To further substantiate this hypothesis, monitoring fluctuations in the dynamics of complexes formed by the OCT1 transporter with specific ligands is crucial for gaining deeper insights into protein and docked complexes under biological conditions. The 100 ns simulations started from the best docking positions as detailed earlier, with base trajectory parameters for each ligand shown in Figure 10. To draw more accurate conclusions regarding the transportation mechanism, elementary descriptive statistics were gathered in Appendix A. The Root Mean Square Deviation (RMSD) measures the average distance between the atoms of a simulated structure and a reference structure over time, providing insight into the overall structural stability. Figure 10a illustrates that this value is fairly stable for the studied systems, except for the last 5 ns of 2′-methylthiamine simulation, where the RMSD noticeably increases. Thiamine and oxythiamine destabilize the transporter structure more strongly than 2′-methylthiamine, as confirmed by the statistics in Appendix A, showing a higher standard deviation and coefficient of variation for 2′-methylthiamine and oxythiamine than for thiamine. The noticeable increase in RMSD for 2′-methylthiamine in the last phase of the simulation could imply an intensified interaction with the receptor, potentially due to steric obstacles along the trajectory. This suggests that RMSD analysis does not rule out the possibility of mechanical blockage for 2′-methylthiamine. Additionally, for the OCT1–thiamine complex, it indicates a higher degree of unrestricted movement compared to 2′-methylthiamine.

SASA (Solvent Accessible Surface Area) measures the surface area of a biomolecule accessible to a solvent, providing insights into specific regions’ accessibility to solvent molecules, which is essential for understanding binding interactions. Changes in SASA values may indicate conformational changes or alterations in the molecule’s environment during simulations or between different conformations. As depicted in Figure 10b, the ligands fall into the following two categories: thiamine and oxythiamine increase SASA compared to the apoform, while 2′-methylthiamine decreases it. A comparison of standard deviations reveals similar instability levels for thiamine and 2′-methylthiamine, while oxythiamine exhibits notably greater stability with a smaller standard deviation than the apoform. Changes in the SASA during the simulation vary, with 2′-methylthiamine showing an increase of around 30 ns, unlike the other systems. These findings support the hypothesis of a stronger blocking by 2′-methylthiamine of OCT1 protein folding, thereby impairing its transport functions.

The radius of gyration (Rg) measures the overall size and compactness of a biomolecular structure, providing insights into the protein folding state and conformational dynamics. Calculated as the root mean square distance of atoms from their common center of mass, monitoring changes in Rg helps us understand structural variability and compactness over time. Initially, the presence of the ligand has little impact on Rg for approximately the first 50 nanoseconds, as depicted in Figure 10c. However, beyond this point, the Rg of the complex with 2′-methylthiamine notably decreases, indicating protein contraction and volume reduction, potentially interfering with its cation transporter function. Such a decrease is not observed for the other systems, indicating the specificity of this derivative.

By measuring the deviation of each residue’s position from its average during the simulation, the Root Mean Square Fluctuation (RMSF) reveals residue flexibility and stability within a protein structure. Figure 10d illustrates RMSF variability for the systems studied. The average values (Appendix A) show minimal deviation from the apoform for thiamine and oxythiamine of about 1.4–1.5 Å. However, the complex with 2′-methylthiamine exhibits a notable decrease (1.25 Å), indicating increased protein folding. Notably, RMSF for residue 470 (Serine), which interacts directly with the ligand via a hydrogen bond, is the lowest for 2′-methylthiamine (0.57 Å). This result suggests decreased ligand mobility, which is consistent with previous investigations. In summary, this analysis highlights ligand-induced changes in residue dynamics with decreased RMSF, indicating a more stable protein–ligand complex, as observed with 2′-methylthiamine.

Apart from the SASA for the entire protein–ligand complex, its per ligand value is also pertinent to transportation abilities. The surface area accessible to the solvent serves as an indicator of the ligand’s exposure to the environment. A higher SASA may suggest increased movement through the environment or interactions with other molecules, potentially facilitating diffusion through cell membranes or other biological structures. For this reason, it is reasonable to suspect that in the cases considered here, the average value of SASA will be the lowest for the complex with 2′-methylthiamine. Indeed, as can be seen from Figure 11, the 2′-methylthiamine ligand exhibits the smallest SASA for most of the simulation, which is also confirmed by its lowest average.

For a deeper understanding of the ligand transportation mechanism, the polar interaction ability of the investigated ligands was assessed by quantifying the number of hydrogen bonds between OCT1 and the ligands over a 100 ns simulation period. Emphasizing the crucial role of hydrogen bonds in mediating molecular recognition and binding, these results provide insights into the potential bonding and capabilities of the transporter protein. Dynamic interactions are not fundamentally different from static ones. The main contacts identified involve serine, particularly SER-470 (for thiamine and 2′-methylthiamine), and tyrosine (THR-245) for oxythiamine. Thiamine and 2′-methylthiamine consistently interacted with SER-470, forming hydrogen bonds exclusively with this residue at the start of the MD simulations. In contrast, oxythiamine showed a different pattern, with interactions constituting only a small percentage (less than 1%) of all contacts. As seen from Figure 8 and Figure 9, the simulation’s starting points differ for oxythiamine compared to thiamine and 2′-methylthiamine, leading to noticeable trajectory differences. The significant involvement of SER-470 binding supports the assertion of slower movement of 2′-methylthiamine by OCT1 compared to other inhibitors. According to Table 6, oxythiamine transport appears to be the most efficient.

## 3. Materials and Methods

### 3.1. General Information on the Synthesis of 2′-Methylthiamine

Acetamidine hydrochloride, sodium ethoxide solution, Raney Nickel, and NaBH_4_ were purchased from commercial suppliers and used as received. The reactions were carried out under an argon atmosphere. 2,4-Dimethyl-5-(2-hydroxyethyl)thiazole was prepared according to the described procedure [27]. All the solvents were used after fractional distillation. IR spectra were acquired on Nicolet 6700 with a Smart Orbit pickup spectrophotometer using ATR (ν cm^−1^). NMR experiments were performed in a Bruker Advance 400 spectrometer. ^1^H and ^13^C NMR chemical shifts (δ) are reported in parts per million (ppm), and they are relative to TMS (0.0 ppm). The data are reported as follows: the chemical shift (the number of hydrogen atoms, multiplicity, and coupling constants, where applicable). The abbreviations are as follows: s (singlet), d (doublet), t (triplet), m (multiplet). The coupling constant (*J*) is quoted in Hz to the nearest 0.1 Hz. The spectra of the obtained compounds (S1–S10) are presented in the Appendix A.

#### 3.1.1. 4-Amino-2-Methylpyrimidine-5-Carbonitrile (**1**) [56]

To the solution of 1 g acetamidine hydrochloride (10.58 mmol) in 20 mL of absolute ethanol, 3.95 mL of sodium ethoxide (21% solution) was added and the reaction mixture was stirred for 1 h at room temperature, during which time the solution turned turbid white. After filtration, 1.343 g of the ethoxymethylenemalononitrile (10.58 mmol) was added in portions and the obtained mixture was stirred overnight. Thereafter, the solvent was evaporated, and the obtained residue was dissolved in glacial acetic acid and cooled. Following this, ammonium solution was added until it became neutral. The precipitated product was filtered, washed with copious amounts of cold water, and then dried to yield 1.007 g (68%) of a yellow solid. Mp = 236–237 °C lit. 246–248 °C (crist EtOH); IR (ATR) ν max cm^−1^ 3377, 3334, 2221, 1673; ^1^H NMR (400 MHz, CD_3_OD) δ/ppm: 8.41 (s, 1H), 2.46 (s, 3H), ^13^C NMR (100 MHz, CD_3_OD) δ/ppm: 171.8, 164.4, 161.3, 115.7, 88.9, 25.9.

#### 3.1.2. 4-Amino-2-Methylpyrimidine-5-Carbaldehyde (**2**) [57]

Raney Nickel (0.25 g of 50% slurry with water) was added to a solution of **1** (0.34 g, 2.5 mmol) in 80% formic acid (2 mL) and the reaction mixture was refluxed for 2 h. Next, the reaction mixture was filtered and washed with 10 mL formic acid. The filtrate and washings were collected and concentrated under a reduced pressure. The resulting residue was purified by column chromatography (dichloromethane–methanol, 50:1 *v*/*v*) to obtain 202 mg (59%) of **2** as an off-white solid. Mp = 198–204 °C; IR (ATR) ν max cm^−1^ 3373, 3133, 1677; ^1^H NMR (400 MHz, CD_3_OD + CDCl_3_) δ/ppm: 9.79 (s, 1H), 8.49 (s, 1H), 2.49 (s, 3H); ^13^C NMR (100 MHz, CD_3_OD + CDCl_3_) δ/ppm: 191.3, 171.6, 163.1, 161.1, 110.2, 25.7.

#### 3.1.3. (4-Amino-2-Methylpyrimidin-5-yl)Methanol (**3**) [58]

NaBH_4_ (0.084 g, 2.21 mmol) was added to a cold methanolic solution of **2** (0.202 g, 1.48 mmol) and the reaction mixture was stirred at room temperature for 3.5 h. After which, the clear reaction mixture was concentrated to dryness and a pale white residue was obtained. After stirring in an additional 3 mL of cold water, the precipitate was observed and subsequently refrigerated for 12 h to obtain the solid. Following filtration and washing with 2 mL of cold water, the solution was dried to obtain 0.185 g (90%) of **3** as a white solid. Mp = 193–195 °C; IR (ATR) ν max cm^−1^ 3362, 3151, 1662, 1421; ^1^H NMR (400 MHz, CD_3_OD) δ/ppm: 7.95 (s, 1H), 4.49 (s, 1H), 2.40 (s, 3H); ^13^C NMR (100 MHz, CD_3_OD) δ/ppm: 166.1, 162.4, 152.1, 113.8, 58.4, 23.4.

#### 3.1.4. 2-[3-[(4-Amino-2-Methylpyrimidin-5-yl)methyl]-2,4-dimethyl-1,3-thiazol-3-ium-5-yl]ethanol (2′-Methylthiamine)

To the solution of 100 mg of **3** (0.72 mmol) in glacial acetic acid, 1.55 mL of HBr (30% in CH_3_COOH) was dropped, and white precipitate appeared. The mixture was then stirred at 60 °C until the starting material disappeared. After azeotropic evaporation to dryness, the residue was dissolved in excess of 2,4-dimethyl-5-(2-hydroxyethyl)thiazole (**4**) in an argon atmosphere. The reaction was performed for 24 h at 100 °C. After that time, the reaction mixture was cooled to room temperature and chromatographic purification was performed (methanol–water, 100:2 *v*/*v*). The crude product was precipitated by treating it with diethyl ether to obtain 63 mg of 2′-methylthiamine as a white solid (20%). Mp = 211–214 °C; IR (ATR) ν max cm^−1^ 3314, 3118, 3028, 1653, 1598; ^1^H NMR (400 MHz, D_2_O) δ/ppm: 7.32 (s, 1H), 5.51 (s, 2H), 3,92 (t, *J* = 5.6 Hz, 2H), 3.19 (t, *J* = 5.6 Hz, 2H), 2.97 (s, 3H), 2.61 (s, 3H), 2.45 (s, 3H); ^13^C NMR (100 MHz, D_2_O) δ/ppm: 173.5, 166.1, 164.4, 144.8, 143.3, 135.5, 110.5, 63.0, 48.7, 31.7, 24.0, 18.2, 14.1; ESI-HRMS *m*/*z*: calcd for [M-HBr-Br^-^]^2+^ 279.1274, found 279.1279.

### 3.2. In Vitro Cell Culture and Cytotoxicity Test

We investigated the cytotoxicity and effects of oxythiamine and 2′-methylthiamine on fibroblast (ATCC-CRL-2106) and HeLa (the human cervical cancer cell line; CCL-2; ATCC, Manassas, VA, USA) cell growth. Fibroblasts were at passage 5 and HeLa cells were at passage 15. Cells were cultured under the 5% CO_2_ and 95% humidity in Medium199 (M4530; Sigma-Aldrich, St. Louis, MO, USA) with 10% fetal bovine serum (F7524; Sigma-Aldrich), 50 U/mL penicillin, and 50 µg/mL streptomycin (P0781; Sigma-Aldrich) at 37 °C. The media was changed every 2 days. Cells were seeded at the density of 1 × 10^5^ cells/well in 12-well plates one day prior to the addition of the test compounds. The control (without antivitamins) and experimental cultures (thiamine, oxythiamine, or 2′-methylthiamine at the concentration of 6–1500 µM) were maintained until 95–100% confluence of the control cultures was achieved (approximately 3 days for HeLa cells and 6 days for fibroblasts). The toxicity of oxythiamine and 2′-methylthiamine was determined by means of colorimetric detection using the MTT test [59]. The cells were incubated for 0.5 h in 0.5 mL of PBS with 50 μL of MTT (5 mg/mL). The medium was then removed from the wells, and formazan crystals were dissolved in 0.5 mL of dimethyl sulfoxide (DMSO) with 0.01 mL of Sorensen’s buffer. Absorbance was measured by means of the Lambda E plate reader (MWG Biotech AG, Ebensburg, Germany) at a wavelength of 570 nm. The results were expressed as a percentage of the controls. In order to assess the effect of the tested thiamine antivitamins on the cell growth, cells were counted in an automated EVE-MT cell counter (NanoEnTec Inc., Seoul, Republic of Korea), and the growth curves of the cultures were plotted. Based on the results of the MTT test and cell counting, the following values have been determined: the Gi50 (growth inhibition—concentration of the tested compound that reduces the number of cells by half as compared to the control), the IC50 (concentration of the tested compound that reduces the cell metabolism rate by half as compared to the control), and the IS (the selectivity index defined as the ratio of cytotoxicity of the compound in relation to normal cells versus cancer cells using the results of the MTT test, according to Cui et al. 2019 [60]). The experiments were conducted in six independent replicates for the purpose of statistical evaluation.

### 3.3. Molecular Docking

The molecular docking studies were conducted to better understand the mechanism of interaction of the proposed derivatives with the enzymes and transporters using the latest (1.2.5) version of the AutoDock Vina program [61]. The Pymol version 2.2.3 [62] and BIOVIA Discovery Studio Visualizer version 21.1.0 [63] packages were employed for the visualization of the results. The structures of human pyruvate dehydrogenase (PDB:3EXF, resolution 3.00 Å [64]), human transketolase (PDB:3MOS, resolution 1.75 Å [65]), mouse thiamine pyrophosphokinase (PDB:2F17, resolution 2.50 Å [29]), the S-component for thiamine from an ECF-type ABC transporter (PDB:3RLB, resolution 2.00 Å [66]), and the human Cryo-EM structure of the organic cation transporter 1 (OCT1, PDB:8ET8, resolution 3.45 Å [67]) were taken from the Protein Data Bank. The primary criterion for selecting these structures was their origin from humans, with further consideration given to factors such as resolution and publication date. Among the available structures, we attempted to choose the most recent one with the highest available resolution. The model, prepared for the docking purposes, underwent pre-treatment including the removal of embedded ligands, solvent, and ions, along with the addition of polar hydrogen atoms and the integration of Kollman charges [68]. A cubic box of 20 Å per edge was established to encompass the ligands during the docking process. In order to ensure optimal accuracy by minimizing the impact of randomness on the results, the simulations were conducted with the parameter “EXHAUSTIVENESS” set to equal 100, surpassing the default value of 8. The objective was to identify the optimal site for inserting the ligand into the proteins. In customary practice, reference compounds were integrated into crystallographic structures, with their placement indicating the location of the active center. However, in the absence of crystallographic data, the presumed position of the active site was unknown. In order to account for variations in the interaction mechanisms of the ligands with the individual receptors, the analysis encompassed the entire receptor area rather than focusing on a specific fragment. This approach proved especially beneficial when crystallographic data were lacking, as observed in scenarios such as in the case of the OCT1 transporter. Consequently, docking calculations were executed using a set of grid boxes overlapping and encompassing the entire range of variations in protein features. The box moved across the enzyme/transporter area. The poses with the best scores were automatically identified through an in-house script, available from the authors. That process was repeated across subsequent grid boxes, thus ensuring the capture of the optimal ligand–receptor arrangement for all the systems under consideration. For the sake of validation of the docking procedure, a redocking process was carried out. The geometries of the resulting poses with the highest scores were compared with that of the TPP from crystal structure 3EXF. As depicted in Figure 12, both conformations show a commendable level of agreement with RMSD of 1790 Å.

### 3.4. Molecular Dynamics

The simulations, conducted using the NAMD/VMD codes [69,70], were initiated with the top poses identified in the docking experiments. The CHARMM22 force field [71] for proteins, augmented with CMAP corrections [72], was utilized. Individual parameter files for the ligands were generated using the Ligand Reader and Modeler tool accessible via the CHARMM-GUI online environment (https://charmm-gui.org/?doc=input/ligandrm, accessed on 10 February 2024) [73]. Water molecules and Na^+^/Cl^−^ ions were introduced into the system. Periodic boundary conditions were applied, with the cell size exceeding that of the macromolecule (OCT1) by 10 Å. To enhance the stability and convergence during the production stage, the simulation began with an initial minimization spanning 50,000 steps. Subsequently, a gradual heating from 0 to 310 K in 2 K increments was undertaken. Following this, an equilibration phase consisting of 200,000 steps was carried out. This phase was crucial to ensure that the system initiated the dynamics from a reasonable starting point. The initial minimization facilitated the relaxation of the system and the more stable atomic arrangement, while mitigating close contacts. The stabilized structures underwent a 100-nanosecond production run with a timestep of 2 femtoseconds, representing a robust approach for studying molecular dynamics and gaining valuable insights into system behavior [74]. The practice of saving frames every 50,000 steps is standard, aiding in various essential tasks such as data management, analysis, simulation restarts, validation, reproducibility, and visualization. Trajectory parameters were computed using scripts provided by the VMD developers. To preserve a constant pressure of 1 atmosphere, the Langevin piston method was employed with a decay period of 100 femtoseconds, chosen to accurately simulate pressure fluctuations in real systems, thereby enhancing the fidelity of the simulations and enabling more realistic dynamic behavior to be observed. Additionally, constant temperature was maintained using Langevin dynamics.

### 3.5. Statistical Analysis

The data from six independent experiments were used for the purpose of statistical calculations using the Statistica 13.0 software (StatSoft; Tulsa, OK, USA). The data were assessed by means of the Shapiro–Wilk W test to check the normality of the data distribution and Levene’s L test to check the homoscedasticity of variance. Where the data had a normal distribution and homoscedastic variance, the differences between means were tested using Student’s *t*-test to compare means. Where the data were not normally distributed, the significance of differences was determined using the Mann–Whitney U or Kruskal–Wallis test at the significance level of *p* < 0.01 to compare medians.

## 4. Conclusions

In the study of cancer cell lines, we have found a significantly stronger inhibition of the culture growth by oxythiamine as compared to 2′-methylthiamine. However, 2′-methylthiamine has the greater specificity towards cancer cells. Both tested derivatives did not significantly affect the fibroblast growth. Summarizing the results obtained, the potential we have indicated for thiamine pyrophosphokinase inhibition by 2′-methylthiamine is such that the use of it as a cytostatic should not be categorically rejected despite the weak effect on cancer cells noted in our experiment. In silico static and dynamic computations provide a mechanistic explanation for that phenomenon. The molecular docking simulations uncover additional non-covalent interactions, as compared to thiamine, which impede the movement of 2′-methylthiamine within the transporter. The dynamic calculations also suggest that the presence of an additional methyl group in the molecule creates mechanistic blockages that decelerate the transport of that ligand. As such, solving the problem of 2′-methylthiamine transport into the cell can cause the inhibition of thiamine pyrophosphokinase activity. Reducing the availability of thiamine pyrophosphate will result in the inhibition of transketolase as well as mitochondrial 2-oxoacid dehydrogenase complexes’ activity, triggering the cytotoxic effect of 2’methylthiamine. Therefore, further studies of this interesting thiamine analogue should focus on finding alternative ways to introduce the aforementioned derivative into cells. For this purpose, liposomes, polymer-carriers, or conjugates of 2′-methylthiamine with compounds actively taken up by cancer cells, such as sugars or folic acid, could be proposed [75,76].

## Data Availability

Data is contained within the article and Appendix A.

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
