# Peer review of "In Vitro and In Silico Studies on Cytotoxic Properties of Oxythiamine and 2′-Methylthiamine"

_ijms, 2024, doi:10.3390/ijms25084359_

Round 1

Reviewer 1 Report

Comments and Suggestions for Authors

Dear authors, 

Thank you for possibility to review your manuscript. Hope my comments will help you to improve the quality of the manuscript:

1) Fig. 3 - in my opinion concentration range 0-10 mkg/ml should be presented in higher resolution - currently IC50 are not clear, especially with standard deviation presented (about 50% of total cell number even with 6 replicates); I suggest to present theses results as a standard survival curves covering 0-100mkg/ml concentration range relative to untreated control (y-axis should represent cell number in % relative to untreated control) with clear plateau (lack of effect concentrations and maximal cell growth inhibition concentrations).

2) Fig 3, Fig 4 and Table 1 generally contain the same information, and in my opinion could be grouped in one plot or figure.

3) It is not enough clearly discussed - why the authors observed toxicity of OT and MT exclusively towards HeLa cells - are there significant differences in expression levels of OCT1 and ThiT transporters in fibroblasts and Hela cells; it would be great to observe the predicted effects applying appropriate negative and positive in vitro models (in further papers).

Comments on the Quality of English Language

Generally English is well, but the authors should thoroughly proofread and corrected the manuscript - I have met plenty of passives (they are accepted only in methods description section), buried predicates, expression faults, infinitives at the start of sentences, and some tense misuse during results description (In my opinion past simple is appropriate in this case for clarity sake).

Author Response

See the file attached.

Reviewer 2 Report

Comments and Suggestions for Authors

The manuscript examines the cytostatic potential of 2'-methylthiamine compared to oxythiamine. Although oxythiamine proves more effective in inhibiting the growth and metabolism of cancer cells, 2'-methylthiamine demonstrates greater selectivity for these cells. Computational analyses reveal discrepancies in the compound's affinity for specific enzymes. Challenges associated with the cellular transport of 2'-methylthiamine may limit its effectiveness, suggesting the need for research to enhance transport and optimize its cytostatic properties.

In this way, I would like to provide some constructive feedback to improve the quality of the manuscript. Below, you will find detailed suggestions and recommendations:

Abstract

Provide a brief introduction to contextualize the problem or significance of the study.

If possible, outline in a simplified manner the methodology used in the study.

Include the clinical implications of the results, suggesting directions for future research, particularly in enhancing the transport of 2'-methylthiamine.

Reinforce the conclusions, emphasizing the significance of the findings and providing a clear outline of the subsequent steps in the research.

Introduction

·         Emphasize a more in-depth exploration of the computational approach in the text, highlighting the use of computational chemistry tools to propose a mechanism of action for 2'-methylthiamine.

·         It would be interesting to conclude the introduction by emphasizing that the research aims to provide a preliminary assessment of the potential use of 2'-methylthiamine as a cytostatic, underscoring its significance in expanding the understanding of thiamine-derived compounds and their possible applications.

Result and discussion

2.2. The effects of oxythiamine and 2’-methylthiamine on cells in vitro

·         Provide greater clarity in distinguishing between the observed results and the proposed mechanism of action for thiamine antimetabolites. Additionally, delve deeper into explaining the relationship between the molecular structure and the inhibitory activity on the cellular growth of the investigated compounds. This structure-activity approach can be introduced at this point and further expanded upon in docking and/or molecular dynamics studies.

·         Relate the selectivity against cancer cells in the study, emphasizing its potential clinical implications.

·         When discussing the results, it is important to draw comparisons with literature data whenever possible and include relevant references. This will enhance the credibility of your conclusions.

·         Highlight the unique contribution that the study offers to understanding the effect of these thiamine antimetabolites on cancer cells..

2.3. Molecular Docking

·         The text would be clearer if the results and discussion were divided into subsections, making the reading more organized and enhancing understanding.

·         Enhance the discussion of computational findings with the in vitro results observed in the study.

·         Review the text and ensure that the language used is clear and accessible.

·         Revise all references to ensure proper citation.

·         Based on the results of the computational study, assess the possibility of suggesting new studies that could be explored in future research.

2.3. Molecular Dynamics

·         Sometimes, the sentences become overly long; therefore, break them into shorter and more straightforward structures to improve readability.

·         Review the text to eliminate redundant sentences or repetitive language to make the text more concise.

3. Materials and Methods

3.2. In vitro cell culture and cytotoxicity test

·         Provide more details about the initial conditions of the cells, such as the number of passages, the culture duration, and the conditions before treatment.

·         Verify whether the reference to Cui et al. 2019 [51] is properly cited in the manuscript.

3.3. Statistical analysis

It is suggested to reorganize the "Statistical Analysis" topic so that it becomes the last point addressed in the methodology section.

3.4. Molecular docking

·         Provide more information about the selection criteria for the chosen targets (PDB). Explain why these specific structures were selected and how they relate to the study's objectives.

3.5. Molecular Dynamics

·         Be clearer in justifying the choice of specific tools, force fields, and methods, highlighting benefits and reasons for selection.

·         Clearly and concisely explain why the chosen cell size exceeds the size of the macromolecule by 10 Å.

·         Ensure appropriate and up-to-date references for the mentioned methods, tools, and techniques.

4. Conclusions

·         Although the text is informative and well-structured, I suggest avoiding references in the conclusion, keeping it focused on the findings and practical implications of the results. The conclusion can be more straightforward in stating the practical implications and emphasizing the need for future research on alternative methods of introducing 2’-methylthiamine into cells.

Author Response

See the file attached.

Reviewer 3 Report

Comments and Suggestions for Authors

The manuscript deals with in vitro and in silico investigations of cytotoxicity of oxythiamine and 2’-methylthiamine. The authors described a synthesis of these compounds and evaluated their effect on normal skin fibroblasts and HeLa cancer cells. It was found that the compounds inhibit the growth and metabolism of cancer cells in the micromolar concentration range. Also authors proposed three possible molecular targets and studied the action of the obtained compounds using the methods of molecular docking. In view that the mechanism of influence vitamin B1 and its derivatives on the proliferation of cancer cell lines is currently not quite clear, the manuscript may be of interest to the scientific community.

Remarks:

1)      The counterion should be added to the formulas of the compounds shown in the fig. 1

2)      The authors wrote (lines 87-88): “Our research aims to synthesize 2’-methylthiamine by a new, simplified method from commercially available substrates”, however, the identical synthetic sequence is described, for example, in refs 9, 21, 22. So, novelty and simplicity of the method is questionable.

3)      The references to described compounds should be given in Materials and Methods

4)      Why HeLa cancer cell line was chosen for a study? Justification is required

5)      GI50 values should be recalculated to µM.

6)      Positive control would be useful in the MTT test

7)      It is known that high-dose vitamin B1 also reduces cancer cell proliferation [10.1007/s00280-014-2386-z]. Taking into account this fact it is of interest to add the comparative analysis of vitamin B1 and the obtained data for oxythiamine and 2’-methylthiamine.

8)      The authors mention (lines 405-406): “Despite that to date we have not found any information about oxythiamine interaction with bacteria.” However, such investigations are described, for example, in the manuscript 10.1016/j.chembiol.2022.07.001

9)      For ref. 23 the name of the journal, year, volume, pages should be given

10)  The ref.67 (line 736, Conclusions) is absent in the list of references.

Author Response

See the file attached.

Reviewer 4 Report

Comments and Suggestions for Authors

Interesting results of your research. Of course I have some issues, though minor.

The accuracy in presenting the measured values in Table 1 does not improve the readability. 

In Fig 2 concentration is given in microM. In the rest of the manuscript concentrations are given in micro g/ml. Better keep it identical through the manuscript

line 165: I50 = IC50?

line 616: IUPAC name should be given like for the intermediates

I could not find anything about the purity of the methylthiamine and 1H-NMR of this compound is not completely clean. Purity should be assessed by HPLC on 2 different wavelengths.

Author Response

See the file attached.

Round 2

Reviewer 1 Report

Comments and Suggestions for Authors

The authors have corrected all the issues I mentioned in previous review round:

1. The results presentation is corrected;

2. English is proofread.